# Vitamin D_3_ Modulates Inflammatory and Antimicrobial Responses in Oral Epithelial Cells Exposed to Periodontitis-Associated Bacteria

**DOI:** 10.3390/ijms26147001

**Published:** 2025-07-21

**Authors:** Fadime Karaca, Susanne Bloch, Fabian L. Kendlbacher, Christian Behm, Christina Schäffer, Oleh Andrukhov

**Affiliations:** 1Competence Center for Periodontal Research, University Clinic of Dentistry, Medical University of Vienna, 1090 Vienna, Austria; fadime.karaca@meduniwien.ac.at (F.K.); susanne.bloch@meduniwien.ac.at (S.B.); christian.behm@meduniwien.ac.at (C.B.); 2*NanoGlycobiology* Research Group, Institute of Biochemistry, Department of Natural Sciences and Sustainable Resources, University of Natural Resources and Life Sciences, 1190 Vienna, Austria; fabian.kendlbacher@boku.ac.at

**Keywords:** oral epithelium, vitamin D_3_, antimicrobial peptides, antimicrobial activity, inflammation, *Tannerella forsythia*, *Fusobacterium nucleatum*, *Porphyromonas gingivalis*

## Abstract

The oral epithelium is essential for maintaining oral health and plays a key role in the onset and progression of periodontitis. It serves as both a mechanical and immunological barrier and possesses antimicrobial activity. Vitamin D_3_, a hormone with known immunomodulatory functions, may influence oral epithelial responses. This study investigated the effects of two vitamin D_3_ metabolites on key immunological and antimicrobial functions of oral epithelial cells, both under basal conditions and during bacterial challenge. Ca9-22 oral epithelial cells were treated with 1,25(OH)_2_D_3_ or 25(OH)D_3_ in the presence or absence of *Tannerella forsythia*, *Fusobacterium nucleatum*, or *Porphyromonas gingivalis*. Inflammatory responses were assessed by measuring gene and protein expression of IL-1β and IL-8. Antimicrobial activity was evaluated via expression of LL-37, hBD-2, and hBD-3, as well as direct bacterial killing assays. Expression of epithelial integrity markers E-cadherin and ICAM-1 was also analyzed. Vitamin D_3_ metabolites reduced IL-8 expression and significantly increased LL-37 expression and production in Ca9-22 cells. Both forms enhanced antimicrobial activity against all tested pathogens and modulated epithelial integrity markers. Vitamin D_3_ positively regulates antimicrobial and barrier functions in oral epithelial cells, suggesting a potential role in supporting oral health and preventing periodontitis progression.

## 1. Introduction

The oral cavity is a complex and dynamic environment where the maintenance of a balanced host–microbiota interaction is essential for oral health [1]. In healthy individuals, the oral microbiome forms a symbiotic relationship with the host’s immune system, sustaining a stable and mutually beneficial ecosystem [2,3,4,5]. Disruption of this homeostasis, often due to microbial dysbiosis, can lead to periodontitis—a prevalent inflammatory disease affecting the tooth-supporting tissues. Classically associated with members of the “red complex”, such as *Porphyromonas gingivalis*, *Tannerella forsythia*, and *Treponema denticola* [6], periodontitis is now understood as a polymicrobial infection involving a broader shift in microbial composition [7]. Notably, an increased abundance of genera such as *Fusobacterium*, *Filifactor*, *Prevotella*, *Tannerella*, and *Porphyromonas*, accompanied by a concurrent reduction in commensal genera such as *Streptococcus*, *Actinomyces*, and *Haemophilus* in subgingival biofilms, has been associated with the onset and progression of periodontal disease [8,9]. This microbial shift suggests a potential role for these genera in the pathogenesis of the disease.

While the pathogenesis of periodontitis is multifactorial and still incompletely understood [10], vitamin D_3_ deficiency has emerged as a potential risk factor [11]. Vitamin D_3_ is a fat-soluble secosteroid hormone synthesized in the skin upon UVB exposure or obtained via diet [12]. It undergoes two-step hydroxylation—first to 25-hydroxyvitamin D_3_ [25(OH)D_3_] in the liver, and then to its active form 1,25-dihydroxyvitamin D_3_ [1,25(OH)_2_D_3_] in the kidneys [13]. The biological activity of 1,25(OH)_2_D_3_ is mediated by the vitamin D receptor (VDR), which is expressed in various oral cell types [14]. VDR is an intracellular nuclear receptor functioning as a ligand-activated transcription factor which directly regulates gene expression (e.g., antimicrobial peptides), but may also exist in a membrane-associated form that mediates the non-genomic effects of vitamin D_3_ (e.g., calcium influx) [15]. The detailed mechanisms of VDR signalling specifically in oral epithelial cells remain incompletely understood.

Vitamin D_3_ status is commonly assessed by measuring serum 25(OH)D_3_ concentrations, with optimal levels typically ranging between 75 and 125 nmol/L (30–50 ng/mL). Serum levels of 25(OH)D_3_ above 50 nmol/L (20 ng/mL) are generally considered, while levels below 30 nmol/L (12 ng/mL) indicate severe vitamin D_3_ deficiency [16,17]. Salivary concentrations of 25(OH)D_3_ have been reported to be up to 5 times lower than those in serum [18,19]; however, no established reference ranges for salivary levels currently exist. Meta-analyses have shown that serum levels of 25(OH)D_3_ are significantly lower in patients with periodontitis compared to healthy individuals [20]. The mechanisms through which vitamin D_3_ contributes to oral health and modulates the progression of periodontitis are currently under active investigation. In particular, vitamin D_3_ is a key regulator of both bone homeostasis and immune function and has recently been implicated in host antimicrobial defense mechanisms [19,20,21].

The oral epithelium plays a critical role in preserving oral health by acting as both a physical and immunological barrier against microbial invasion [22]; it is in constant and dynamic interaction with the resident microbiota under both healthy and pathological conditions. Numerous studies have investigated how specific periodontitis-associated bacteria interact with epithelial cells, providing key insights into their role in disease progression. Among these, *P. gingivalis* is particularly well studied. This keystone pathogen is capable of adhering to, invading, and replicating within oral epithelial cells, allowing it to evade immune surveillance and establish persistent infections [23]. In addition, *P. gingivalis* can compromise epithelial barrier integrity, facilitating its translocation into deeper connective tissues [24,25]. Its effects on the host immune response are multifaceted, such as upregulating the expression of pro-inflammatory cytokines like IL-1β and IL-8 in oral epithelial cells [26], while simultaneously degrading these mediators through its proteolytic enzymes, thereby subverting host defenses. *T. forsythia* also exhibits strong epithelial tropism, with the ability to adhere to and invade epithelial cells [27,28]. It has also been shown to activate inflammatory signaling pathways, leading to the upregulation of various pro-inflammatory mediators in multiple oral epithelial cell lines [29]. Similarly, *Fusobacterium* can adhere to and invade oral epithelial cells, trigger robust inflammatory responses, and modulate the production of antimicrobial peptides such as defensins and cathelicidins [30,31].

While previous studies have addressed some of these bacterial interactions and the effects of vitamin D_3_ in isolation, little is known about how vitamin D_3_ modulates the epithelial response to bacterial infection. Multiple studies have examined the effects of vitamin D_3_ on oral and gingival epithelial cells, where vitamin D_3_ metabolites modulate critical components of the innate immune response, barrier and antimicrobial functions [32,33,34,35,36,37]. Loss of VDR signaling enhances proliferation and impairs differentiation of oral keratinocytes [32], while 1,25(OH)_2_D_3_ has been shown to promote epithelial cell proliferation and wound closure in vitro [38]. Moreover, 1,25(OH)_2_D_3_ consistently reduces the expression of pro-inflammatory cytokines, including IL-1β and IL-8, in various oral epithelial cells [33,34,35,36,37]. It also upregulates the expression of CD14, and the triggering receptor expressed on myeloid cells-1 (TREM-1), potentially enhancing epithelial innate immunity [39]. Importantly, gingival epithelial cells express the enzyme 1α-hydroxylase (CYP27B1), which catalyzes the local conversion of 25-hydroxyvitamin D_3_ [25(OH)D_3_] to its biologically active form, 1,25-dihydroxyvitamin D_3_ [1,25(OH)_2_D_3_] [33]. Both vitamin D_3_ metabolites are potent inducers of the antimicrobial peptide LL-37 in oral epithelial cells [33,39,40,41], and 1,25(OH)_2_D_3_ additionally increases the expression of human β-defensin 3 (hBD-3) [36]. These effects are functionally relevant, as vitamin D_3_-treated epithelial cells exhibit enhanced antimicrobial activity, particularly against *P. gingivalis* and *Aggregatibacter actinomycetemcomitans* [36,39].

In this study, we investigated how 1,25(OH)_2_D_3_ and 25(OH)D_3_ influence the immune and antimicrobial responses of Ca9-22 oral squamous carcinoma cells in the absence and presence of key periodontitis-associated bacteria: *T. forsythia*, *F. nucleatum*, and *P. gingivalis*. The use of Ca9-22 cells allows overcoming the difficulties in culturing the primary oral epithelial cells [42,43]. We analyzed the expression of inflammatory mediators (IL-1β, IL-8), antimicrobial peptides cathelicidin (LL-37), human β-defensins (hBD-2 and hBD-3), barrier-related molecules (E-cadherin, ICAM-1), as well as the overall antimicrobial activity of vitamin D_3_-treated cells. This study provides new insights into the multifaceted role of vitamin D_3_ in shaping epithelial defense mechanisms and highlights its potential as a modulator of host–microbiome interactions in the context of periodontal disease.

## 2. Results

### 2.1. Production of Pro-Inflammatory Mediators

The effect of 1,25(OH)_2_D_3_ and 25(OH)D_3_ on IL-1β and IL-8 expression and production in Ca9-22 cells, both in the presence and absence of different bacterial infections, is presented in Figure 1.

1,25(OH)_2_D_3_ significantly upregulated IL-1β gene expression in uninfected cells and in those infected with *T. forsythia* (Figure 1A)., whereas no significant changes were observed in cells infected with *P. gingivalis* or *F. nucleatum*. In contrast, 25(OH)D_3_ had no notable effect on IL-1β gene expression in any condition. IL-1β protein levels in the culture supernatants were significantly decreased in *F. nucleatum*-infected cells and significantly increased in *P. gingivalis*-infected cells following treatment with 1,25(OH)_2_D_3_, compared to untreated and 25(OH)D_3_-treated controls (Figure 1B).

Regarding IL-8, both vitamin D_3_ metabolites generally suppressed its gene expression (Figure 1C). This reduction was statistically significant in all groups, except for 25(OH)D_3_-treated cells infected with *T. forsythia* or *P. gingivalis*. At the protein level, a significant decrease in IL-8 secretion was observed only in *F. nucleatum*-infected cells treated with 25(OH)D_3_ (Figure 1D). In all other conditions, neither 1,25(OH)_2_D_3_ nor 25(OH)D_3_ significantly altered IL-8 protein production by Ca9-22 cells.

### 2.2. Production of Antimicrobial Peptides and Antimicrobial Activity

The effects of vitamin D_3_ metabolites on the expression and secretion of cathelicidin antimicrobial peptide (CAMP/LL-37) are presented in Figure 2. Both 1,25(OH)_2_D_3_ and 25(OH)D_3_ significantly upregulated CAMP gene expression in Ca9-22 cells across all tested conditions, including both uninfected and bacterially infected cells (Figure 2A). Notably, 1,25(OH)_2_D_3_ induced higher CAMP mRNA levels than 25(OH)D_3_. At the protein level, LL-37 levels in the conditioned media were also significantly elevated by both metabolites (Figure 2B), although 25(OH)D_3_ led to higher LL-37 peptide concentrations (mean values 14–73 ng/mL) than 1,25(OH)_2_D_3_ (mean values 4–24 ng/mL).

Figure 3 illustrates the effects of vitamin D_3_ metabolites on the expression of human β-defensins in Ca9-22 cells, with and without bacterial infection. Both 1,25(OH)_2_D_3_ and 25(OH)D_3_ significantly downregulated hBD-2 gene expression in uninfected cells (Figure 3A). In infected cells, 25(OH)D_3_ significantly suppressed hBD-2 expression following *T. forsythia* infection, while 1,25(OH)_2_D_3_ significantly decreased hBD-2 expression in *F. nucleatum*-infected cells. Regarding hBD-3, a significant increase in gene expression was observed only in *T. forsythia*-infected cells treated with 1,25(OH)_2_D_3_ (Figure 3B). Moreover, hBD-3 expression was higher in cells treated with 25(OH)D_3_ than those treated with 1,25(OH)_2_D_3_ when infected with *F. nucleatum*.

The antimicrobial activity of Ca9-22 cells treated with vitamin D_3_ metabolites was assessed as described in Section 4.7, and the results are presented in Figure 4. Conditioned media from cells treated with either 1,25(OH)_2_D_3_ or 25(OH)D_3_ significantly inhibited the growth of *T. forsythia*, *F. nucleatum*, and *P. gingivalis* (Figure 4A). A direct antimicrobial effect was also observed when bacteria were exposed to vitamin D_3_ metabolites alone, in the absence of conditioned media (Figure 4B). The percentages of viable bacterial cells following treatment with either conditioned media or vitamin D_3_ metabolites alone are summarized in Figure 4C–E. Notably, conditioned medium from 25(OH) D_3_-treated Ca9-22 cells exhibited a strong antibacterial effect against *T. forsythia* and *P. gingivalis*. In contrast, conditioned medium from 1,25(OH)_2_D_3_-treated cells showed pronounced inhibition only against *P. gingivalis*. In other cases, a slight reduction in bacterial viability was observed following treatment with conditioned media, although these changes did not reach statistical significance.

### 2.3. Gene and Protein Expression of Adhesion Molecules

Figure 5 shows the effects of 1,25(OH)_2_D_3_ and 25(OH)D_3_ on E-cadherin expression at both gene and protein levels in Ca9-22 cells. Treatment with 1,25(OH)_2_D_3_ significantly upregulated E-cadherin gene expression in both uninfected and bacterially infected cells (Figure 5A). In contrast, 25(OH)D_3_ had no significant impact on E-cadherin mRNA levels. At the protein level, flow cytometry analysis revealed minimal changes in the percentage of E-cadherin+ cells. A small but statistically significant decrease was observed with 25(OH)D_3_ in uninfected cells (from 94.2% to 93.3%) (Figure 5B). Interestingly, the mean fluorescence intensity (MFI) of E-cadherin+ cells was significantly reduced by 1,25(OH)_2_D_3_ in the absence of infection. Moreover, in both uninfected cells and cells infected with *T. forsythia* or *P. gingivalis*, the MFI was significantly lower after treatment with 1,25(OH)_2_D_3_ compared to 25(OH)D_3_.

Figure 6 illustrates the impact of vitamin D_3_ metabolites on ICAM-1 gene and protein expression in Ca9-22 cells. ICAM-1 gene expression was significantly decreased only by 1,25(OH)_2_D_3_ in *P. gingivalis*-infected cells (Figure 6A); no other significant changes were observed. The percentage of ICAM-1^+^ cells was significantly reduced by 1,25(OH)_2_D_3_ in uninfected cells (Figure 6B). In *T. forsythia*-infected cells, a trend toward reduction was seen, but no consistent or significant effects were observed for other conditions. Mean fluorescence intensity of ICAM-1^+^ cells was significantly decreased by both vitamin D_3_ metabolites in uninfected cells (Figure 6C). Specifically, 25(OH)D_3_ reduced MFI. in *T. forsythia*-infected cells, and 1,25(OH)_2_D_3_ decreased MFI in *F. nucleatum*-infected cells; however, 1,25(OH)_2_D_3_ significantly increased ICAM-1 MFI in *T. forsythia*-infected cells. When comparing metabolites, 1,25(OH)_2_D_3_-treatment resulted in significantly higher ICAM-1 MFI than 25(OH)D_3_ in both uninfected and *T. forsythia*-infected cells, whereas the opposite was observed in *F. nucleatum*-infected cells. It is worth noting that, despite statistical significance, the changes in ICAM-1^+^ cell percentages and MFI induced by either metabolite were relatively modest—generally not exceeding an 8% difference compared to untreated cells.

## 3. Discussion

The oral epithelium plays a central role in innate immune defense, acting as both a mechanical and immunological barrier against invading pathogens [22]. Vitamin D_3_, a hormone essential for immune regulation, has been implicated in maintaining oral health [11,44]. This study provides a comprehensive analysis of how two key metabolites of vitamin D_3_—1,25(OH)_2_D_3_ (the active form) and 25(OH)D_3_ (the circulating precursor)—modulate the response of oral epithelial cells to periodontitis-associated bacteria. Notably, our results reveal previously unrecognized roles for 25(OH)D_3_ in enhancing antimicrobial responses and gene regulation, independent of its conversion to the active form.

Using Ca9-22 cells, which mimic several features of oral epithelium despite inherent limitations [43,45], we investigated the regulation of inflammatory mediators [46], antimicrobial peptides, barrier-related proteins, and molecules involved in neutrophil migration. We found that 1,25(OH)_2_D_3_ upregulated IL-1β gene expression in the absence of infection and in the presence of *T. forsythia* (Figure 1A), but not at the protein level. This likely reflects the inherently low capacity of Ca9-22 cells to produce IL-1β protein, as levels were near the ELISA-detection threshold. A small but statistically significant increase in IL-1β protein was observed following treatment with 1,25(OH)_2_D_3_ in *F. nucleatum*- and *P. gingivalis*-infected cells (Figure 1B), although the biological relevance is limited due to low baseline levels. These findings contrast with previous work showing inhibitory effects of 1,25(OH)_2_D_3_ on lipopolysaccharide-induced IL-1β expression in human oral keratinocytes [35], possibly due to differences in experimental models. Interestingly, upregulation of IL-1β has been linked to improved barrier integrity in the oral epithelium [47], warranting further investigation into potential barrier-enhancing effects mediated by vitamin D_3_.

Both vitamin D_3_ metabolites suppressed IL-8 gene expression, regardless of bacterial infection, though not all effects reached statistical significance (Figure 1C). At the protein level, only 25(OH)D_3_ significantly reduced IL-8 in *F. nucleatum*-infected cells (Figure 1D). Notably, IL-8 production was minimal in response to *P. gingivalis*, likely due to the proteolytic degradation of chemokines by this pathogen [48,49]. These findings are consistent with studies showing vitamin D_3_-mediated suppression of IL-8 in gingival fibroblasts and periodontal ligament cells [50,51,52], and suggest an anti-inflammatory role through reduced neutrophil recruitment.

Our results also show that both vitamin D_3_ metabolites robustly induced LL-37, a key antimicrobial peptide [53] with anti-inflammatory properties believed to play a role in maintaining oral health [54,55]. This induction was observed at both the gene and protein levels (Figure 2A,B), regardless of bacterial presence. This confirms and extends previous studies showing vitamin D_3_-mediated upregulation of LL-37 in oral epithelial cells [33,39,40,41].

Additionally, we examined the expression of β-defensins [56]. The levels of hBD-2 in the gingival sulcus are increased in periodontitis [57]. hBD-3 was shown to inhibit the inflammatory response in vitro and the development of periodontitis in mice in vivo [58]. Both 1,25(OH)_2_D_3_ and 25(OH)D_3_ reduced hBD-2 expression in uninfected cells, with minimal effects during bacterial infection. Conversely, 25(OH)D_3_ enhanced hBD-3 gene expression, significantly so in *T. forsythia*-infected cells. Expression levels induced by 25(OH)D_3_ were also significantly higher than those induced by 1,25(OH)_2_D_3_. This finding aligns with a previous study showing upregulation of hBD-3 by 1,25(OH)_2_D_3_ [34] and suggests a more pronounced role for 25(OH)D_3_ in modulating defensin responses.

Crucially, both metabolites enhanced the antimicrobial activity of Ca9-22 cells against oral pathogens (Figure 4A), with 25(OH)D_3_ demonstrating greater potency, likely reflecting stronger induction of CAMP and hBD-3. This is the first study to demonstrate such an effect for 25(OH)D_3_, expanding current knowledge beyond its function as a precursor. Direct antimicrobial effects of both metabolites were also observed (Figure 4B), although substantially weaker than cell-mediated effects. Previous studies have demonstrated direct effects of 1,25(OH)_2_D_3_ at substantially higher concentrations [59] compared to the much lower levels used in our study (~3-00 µg/mL vs. ~4 ng/mL). Our findings suggest that even low-dose effects, particularly of 25(OH)D_3_, may be biologically relevant. Interestingly, the antimicrobial efficacy varied among the tested species. Notably, for *F. nucleatum*, a reduced number of viable bacteria was observed even in the control group lacking both host cells and vitamin D (Figure 4B), indicating that the experimental conditions alone may have negatively affected bacterial viability.

We also evaluated E-cadherin, a key molecule in epithelial barrier maintenance, often degraded during bacterial infection [60,61]. While 1,25(OH)_2_D_3_ increased E-cadherin gene expression (Figure 5A), this did not translate into increased surface expression. Instead, a slight decrease in surface protein levels was observed (Figure 5B,C), suggesting a disconnect between transcriptional and post-translational regulation. This contrasts with findings in other epithelial cells [62] and highlights the need for further studies in oral epithelium, particularly given the association between E-cadherin dysregulation and periodontitis [63] and that a potential regulation of E-cadherin by vitamin D_3_ might impact periodontal disease pathogenesis. When interpreting E-cadherin data, it should be noted that E-cadherin expression is diminished in oral squamous carcinoma compared to the normal epithelium [56,64].

Regarding ICAM-1, which mediates trans-epithelial leukocyte migration [65], we observed minor and context-specific changes. 1,25(OH)_2_D_3_ reduced ICAM-1 gene expression in *P. gingivalis*-infected cells (Figure 6A), while changes in surface protein expression were modest and did not exceed 8% compared to untreated cells (Figure 6B,C). Previous studies have demonstrated downregulation of ICAM-1 by vitamin D_3_ in lung epithelial cells [66]; however, the relevance of oral epithelium remains unclear and may depend on the inflammatory microenvironment.

Our findings underscore the complex and context-dependent role of vitamin D_3_ in oral epithelial immunity. Importantly, we demonstrate that 25(OH)D_3_ has independent, immunomodulatory activity at physiologically relevant concentrations (10 nmol/L), comparable to salivary levels [19]. This is in contrast to serum levels of 1,25(OH)_2_D_3_, which are much lower than the concentrations used in vitro [67]. While serum 25(OH)D_3_ levels are a known predictor of periodontitis risk [11], our data suggest that local actions of both vitamin D_3_ metabolites in the epithelium may contribute to this association.

Most oral cells can convert 25(OH)D_3_ to 1,25(OH)_2_D_3_ [44,68], but the physiological relevance of this conversion, particularly in inflamed tissue, remains to be defined. It is also possible that inflammatory conditions suppress vitamin D_3_ signaling [69,70], adding another layer of complexity to its function in periodontal disease. In our study, we observed that 25(OH)D_3_ exerts biological effects in Ca9-22 cells, suggesting that these cells are also capable of converting it into 1,25(OH)_2_D_3_. This still seems possible despite the expression of CYP27B1, an enzyme responsible for the conversion of 25(OH)D3 into 1,25(OH)2D3 is diminished in oral squamous carcinoma [71]. The physiological relevance of local vitamin D_3_ activation, as well as the exact cellular mechanisms of its action, needs to be clarified by further studies.

This study has several limitations. It was conducted in vitro using a squamous carcinoma cell line, which does not fully mimic the behavior of normal oral epithelium. Additionally, the infection model involved a single bacterial species, whereas in vivo exposure typically involves polymicrobial biofilms and complex host-microbe interactions. Moreover, the experimental design did not account for oral commensal bacteria, which play a crucial role in maintaining oral health and the resilience of the subgingival ecological niche to external stimuli. Therefore, investigating the influence of commensals on the effects mediated by vitamin D_3_ represents an essential step towards developing a model that more accurately mimics the in vivo oral ecosystem.

## 4. Materials and Methods

### 4.1. Cell Cultures

The human oral squamous cell carcinoma (OSCC) cell line Ca9-22 was employed as an in vitro model for oral epithelial cells [43]. Cells were maintained in Minimum Essential Medium Eagle (MEM; Sigma, St. Louis, MO, USA) supplemented with 10% fetal bovine serum (FBS; Gibco, Carlsbad, CA, USA), 100 U/mL penicillin, 100 μg/mL streptomycin (P/S; Gibco, Carlsbad, CA, USA) Cultures were incubated at 37 °C in a humidified atmosphere with 5% CO_2_. All experiments were conducted using cells between passages 3 and 7.

### 4.2. Cultivation of Bacteria

*Tannerella forsythia* ATCC 43,037 (American Type Culture Collection, Manassas, VA, USA) was cultured anaerobically at 37°C for 5 days on Fastidious Anaerobe agar (FA; Neogen, Lansing, MI, USA) supplemented with 5% horse blood (Sigma, St.-Louis, MO, USA) and 20 μg/mL *N*-acetylmuramic acid (NAMA; Sigma-Aldrich, St. Louis, MO, USA) [29,72]. *Fusobacterium nucleatum* subsp. *nucleatum* KP-F8 (OMZ598) and *Porphyromonas gingivalis* ATCC 33,277 were grown anaerobically at 37 °C for 5 days on Columbia Blood Agar (CBA; Oxoid, Altrincham, UK) supplemented with 5% horse blood [73].

### 4.3. Infection Procedure

Cells were seeded as a monolayer at a density of 1.7 × 10^5^ cells/well in 500 µL of MEM supplemented with 10% FBS and antibiotics in 24-well plates. After 24 h, the medium was replaced with MEM lacking FBS and antibiotics, and cells were treated with either 10 nmol/L of 1,25(OH)_2_D_3_ or 10 nmol/L 25(OH)D_3_ (both from Cayman Chemical, Ann Arbor, MI, USA). FBS was omitted to avoid potential interference from its components with host-bacteria interactions [74]. Subsequently, cells were infected with either *T. forsythia*, *F. nucleatum*, or *P. gingivalis* at a multiplicity of infection (MOI) of 50. For this purpose, bacterial cells were harvested from agar plates, washed with MEM, and the optical density at 600 nm (OD_600_) was measured using a Biochrom Ultrospec 10 cell density meter (Harvard Bioscience, Holliston, MA, USA). For each bacterial suspension, the correlation between OD_600_ and colony-forming units, as determined previously [48] was used as a basis for calculating the MOI. The concentrations of vitamin D_3_ metabolites and MOI were based on our previously established protocols [49,68,75]. Each treatment and infection combination was performed in duplicate. Cells were exposed to both vitamin D_3_ metabolites and bacteria for 24 h, and gene and protein expression of selected markers was analyzed.

### 4.4. Reverse Transcription-Quantitative Polymerase Chain Reaction (Rt-Qpcr)

Cells were lysed, and mRNA was reverse transcribed to cDNA using a TaqMan Gene Expression Cells-to-Ct kit (Invitrogen, Waltham, MA, USA) following the manufacturer’s protocol, as previously described [49]. cDNA synthesis was carried out on a Biometra TOne PCR thermal cycler (Analytik Jena, Jena, Germany) under the following conditions: 37 °C for 1 h, 95 °C for 5 min, followed by cooling to 4 °C. qPCR was performed to detect specific transcripts using TaqMan Gene Expression Assays (Applied Biosystems, Waltham, MA, USA). The following inventoried assay IDs were used: DEFB103 (β-defensin 103): Hs00218678_m1; IL-1β (interleukin-1β): Hs01555410_m1; IL-8 (interleukin-8): Hs00174103; DEFB4B (β-defensin 4B): Hs00823638_m1; CDH1 (E-cadherin): Hs01023894_m1; ICAM1 (intercellular adhesion molecule 1): Hs00164932_m1; CAMP (cathelicidin antimicrobial peptide, LL-37): Hs00189038_m1; GAPDH (housekeeping gene): Hs99999905_m1. qPCR was conducted using a QuantStudio 3 Real-Time PCR System (Applied Biosystems, Waltham, MA, USA) with the following thermal cycling conditions: 95 °C for 10 min, followed by 50 cycles at 95 °C for 15 s (denaturation) and 60 °C for 1 min (annealing and extension). Cycle threshold values were determined for each gene. The gene expression was quantified in relation to that of GAPDH by the 2^−ΔCt^ method, where ΔC_t_ = (C_t_^target^ − C_t_^GAPDH^).

### 4.5. Protein and Peptide Production Measurement by Elisa

Supernatants were collected at the end of the treatment period and stored at −80 °C until analysis. The concentrations of IL-1β, IL-8, and CAMP were quantified using commercially available ELISA kits, according to the manufacturers’ protocols.: IL-1β (Cat. Nr. 88-7261-88, Invitrogen, Waltham, MA, USA); IL-8 (Cat. Nr. 88-8086-88, Invitrogen, Waltham, MA, USA); CAMP (Cat. Nr. SEC419Hu, Cloud-Clone Corp., Houston, TX, USA). Absorbance was measured using a microplate reader (BioTek Instruments, Winooski, VT, USA), and data were analyzed with Gen5 software (Version 2.09, BioTek Instruments, Winooski, VT, USA).

### 4.6. Analysis of Icam1 and E-Cadherin by Flow Cytometry

At the end of the experiment, Ca9-22 cells from two wells were pooled and transferred into FACS tubes. The culture medium was removed by centrifugation, and the cells were resuspended in 100 µL of FACS buffer (PBS supplemented with 3% BSA and 0.09% NaN_3_). Cells were stained with either PE-conjugated anti-ICAM-1 antibody (Cat. Nr. 12-0549-92, eBioscience, Santa Clara, CA, USA) or PE-conjugated anti-E-cadherin antibody (Cat. Nr. A15784, Life Technologies, Carlsbad, CA, USA), following the manufacturer’s instructions. Immediately after staining, samples were analyzed using an Attune NxT Flow Cytometer (Thermo Fisher Scientific, Waltham, MA, USA). Data were processed using Attune NxT Flow Cytometer Software v3.1, with results reported as mean fluorescence intensity (MFI) and percentage of positive cells relative to the stained untreated control.

### 4.7. Antibacterial Activity of Ca9-22 Cells

Ca9-22 cells were treated with either 10 nmol/L of 1,25(OH)_2_D_3_ or 10 nmol/L 25(OH) D_3_ in the absence of bacterial infection for 24 h, as described above. Following treatment, conditioned media were collected and subsequently applied to bacterial cultures. A total of 500 µL of conditioned media–either untreated or treated with 10 nmol/L of 1,25(OH)_2_D_3_ or 10 nmol/L of 25(OH)D_3_–was mixed with 500 µL of *T. forsythia, F. nucleatum,* or *P. gingivalis* suspended in MEM, at a multiplicity of infection (MOI) of 50. The mixtures were incubated under anaerobic conditions for 24 h, after which 50 µL from each was plated on agar plates for bacterial enumeration. Plating was performed using sterilized solid glass beads (4 mm diameter; (Sigma, St. Louis, MO, USA) to evenly distribute the bacterial suspension. Colony-forming units (CFU) were counted from seven technical replicates. As a control, cell culture medium containing 10 nmol/L of 1,25(OH)_2_D_3_ or 10 nmol/L of 25(OH)D_3_, not pre-conditioned with Ca9-22 cells, was used.

### 4.8. Statistical Analysis

Data are presented as mean values ± standard error of the mean (SEM) from at least five biological replicates. Statistical differences between experimental groups were assessed using the Wilcoxon signed-rank test. Differences in the percentage of viable bacteria between treatments with and without conditioned media were analyzed by the Mann–Whitney U test. Analyses were performed using IBM SPSS 27.0 software (IBM, Chicago, IL, USA). A *p*-value of <0.05 was considered statistically significant. Data visualization was carried out using dot plots generated in GraphPad Prism version 8.0.2. (La Jolla, CA, USA).

## 5. Conclusions

In conclusion, our study is the first to comprehensively show that 25(OH)D_3_, at physiologically relevant concentrations, exerts direct immunomodulatory and antimicrobial effects on oral epithelial cells, complementing and sometimes exceeding the effects of 1,25(OH)_2_D_3_. These findings expand our understanding of vitamin D_3_ biology in the oral cavity and underscore its potential as a modulator of epithelial defense. Further in vivo and clinical studies are needed to determine the translational relevance of these findings.

## Figures and Tables

**Figure 1 ijms-26-07001-f001:**
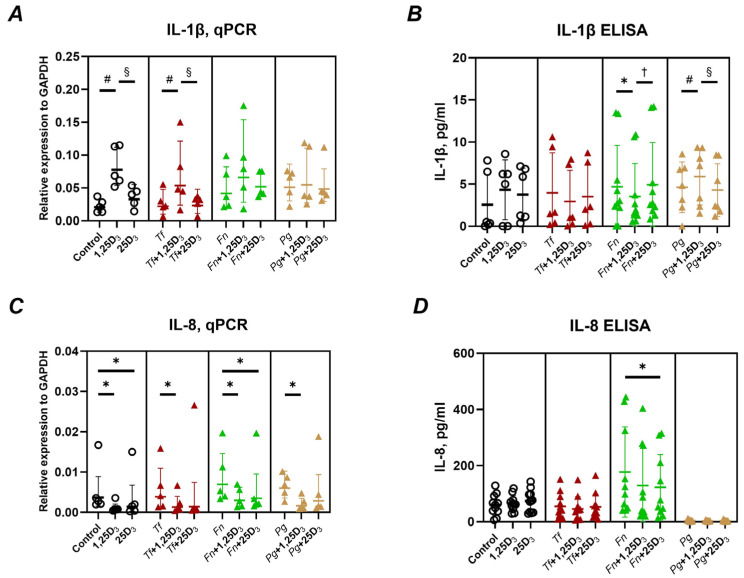
Effect of different vitamin D_3_ metabolites on the gene expression and production of inflammatory mediators in Ca9-22 cells. Ca9-22 cells were treated with 10 nmol/L of either 1,25(OH)_2_D_3_ or 25(OH)D_3_ for 24 h. Treatments were performed in both uninfected cells (circle) and cells infected with *T. forsythia* (red triangles), *F. nucleatum* (green triangles), or *P. gingivalis* (yellow triangles) at an MOI of 50. Following treatment, the gene expression and protein levels of IL-1β (**A**,**B**) and IL-8 (**C**,**D**) were assessed using qPCR and ELISA, respectively. Panels (**A**,**C**) show the gene expression relative to that of GAPDH calculated by the 2^−ΔCt^ method. Panels (**B**,**D**) represent the concentrations of the corresponding proteins in the conditioned media after treatment and infection, respectively. Each data point represents an individual experiment; lines and error bars indicate the mean and SD, respectively. Symbols indicate statistically significant differences (*p* < 0.05). *—significantly reduced gene expression or protein production following vitamin D_3_ metabolite treatment. ^#^—significantly increased gene expression following vitamin D_3_ metabolite treatment. ^§^—significantly increased gene expression after the treatment with 1,25(OH)_2_D_3_ compared to 25(OH)D_3_. ^†^—significantly decreased gene expression after the treatment with 1,25(OH)_2_D_3_ compared to 25(OH)D_3_.

**Figure 2 ijms-26-07001-f002:**
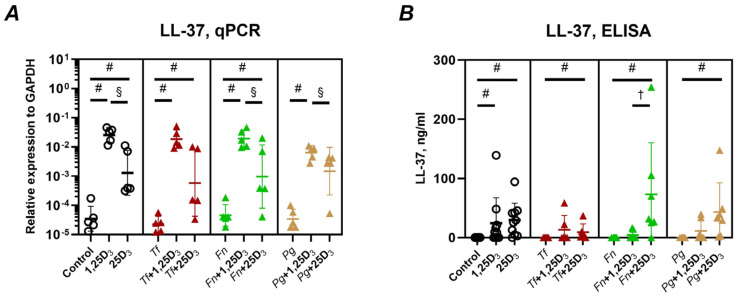
Effect of vitamin D_3_ metabolites on the gene expression and production of cathelicidin antimicrobial peptide LL-37 in Ca9-22 cells. Ca9-22 cells were treated with 10 nmol/L of either 1,25(OH)_2_D_3_ or 25(OH)D_3_ for 24 h. Treatments were applied to both uninfected cells (circle) and cells infected with *T. forsythia* (red triangles), *F. nucleatum* (green triangles), or *P. gingivalis* (yellow triangles) at an MOI of 50. Following treatment, the gene expression and protein production of cathelicidin antimicrobial peptide (CAMP, LL-37) were measured by qPCR and ELISA, respectively. Panel (**A**) shows the gene expression relative to that of GAPDH calculated by the 2^−ΔCt^ method. Panel (**B**) displays the concentration of LL-37 in the conditioned media after treatment. Each data point represents an individual experiment; lines and error bars indicate the mean and SD, respectively. Symbols indicate statistically significant differences (*p* < 0.05). ^#^—significantly increased gene expression after vitamin D_3_ metabolite treatment ^§^—significantly increased gene expression after treatment with 1,25(OH)_2_D_3_ compared to 25(OH)D_3_. ^†^—significantly lower peptide production after treatment with 1,25(OH)_2_D_3_ compared to 25(OH)D_3_.

**Figure 3 ijms-26-07001-f003:**
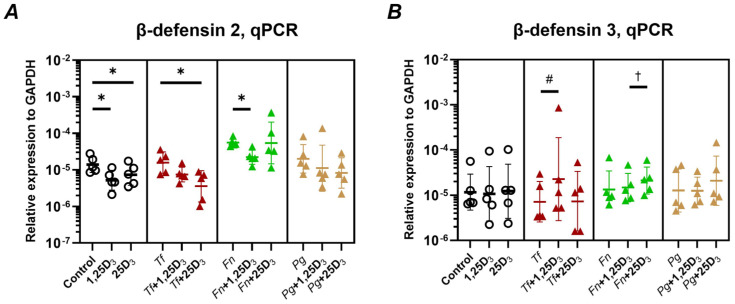
Effect of vitamin D_3_ metabolites on the gene expression and production of β-defensins in Ca9-22 cells. Ca9-22 cells were treated with 10 nmol/L of either 1,25(OH)_2_D_3_ or 25(OH)D_3_ for 24 h. Treatments were applied to both uninfected cells (circle) and cells infected with *T. forsythia* (red triangles), *F. nucleatum* (green triangles), or *P. gingivalis* (yellow triangles) at an MOI of 50. Following treatment, the gene expression of β-defensin 2 (**A**) and β-defensin-3 (**B**) was determined by qPCR. The gene expression relative to that of GAPDH was calculated by the 2^−ΔCt^ method. Each data point represents an individual experiment; lines and error bars indicate the mean and SD, respectively. Symbols indicate statistically significant differences (*p* < 0.05). *—significantly lower gene expression following vitamin D_3_ metabolite treatment. ^#^—significantly higher gene expression following vitamin D_3_ metabolite treatment. ^†^—significantly lower gene expression following the treatment with 1,25(OH)_2_D_3_ compared to 25(OH)D_3_.

**Figure 4 ijms-26-07001-f004:**
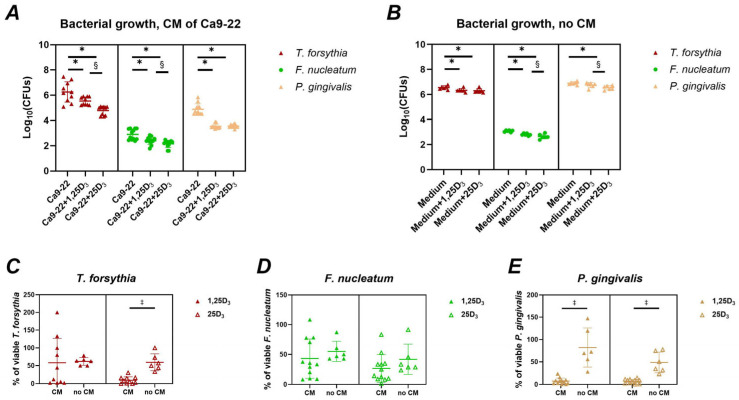
Antimicrobial activity of vitamin D_3_ metabolites mediated by Ca9-22 cells. Panel (**A**) Ca9-22 cells were treated with 10 nmol/L of either 1,25(OH)_2_D_3_ or 25(OH)D_3_ for 24 h. Conditioned media (CM) were then collected and used to treat *T. forsythia* (red triangles), *F. nucleatum* (green triangles) or *P. gingivalis* (yellow triangles). Panel (**B**) Media containing 10 nmol/L of either 1,25(OH)_2_D_3_ or 25(OH)D_3_, but without CA9-22 cells (no CM), was used to treat *T. forsythia*, *F. nucleatum* or *P. gingivalis*. Following treatment, bacteria were plated on selective agar plates and incubated anaerobically for 24 h, followed by counting colony-forming units (CFUs). Panels (**C**–**E**) Percentage of viable cells relative to the corresponding groups not treated with 1,25(OH)_2_D_3_ or 25(OH)D_3_. Each data point represents an individual experiment; lines and error bars indicate the mean and SD, respectively. Symbols denote statistically significant differences (*p* < 0.05). *—significantly lower CFU counts following treatment with vitamin D_3_ metabolites. ^§^—significantly higher CFUs counts after treatment with 1,25(OH)_2_D_3_ or 1,25(OH)_2_D_3_-conditioned media compared to 25(OH)D_3_. ^‡^—significantly different % of live bacteria between treatment with and without CM.

**Figure 5 ijms-26-07001-f005:**
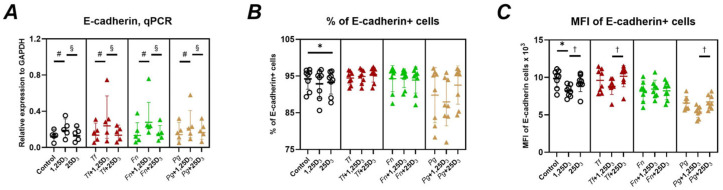
Effect of vitamin D_3_ metabolites on E-cadherin gene and surface protein expression in Ca9-22 cells. Ca9-22 cells were treated with 10nmol/L of either 1,25(OH)_2_D_3_ or 25(OH)D_3_ for 24 h. Treatments were performed on uninfected cells (circle) and cells infected with *T. forsythia* (red triangles), *F. nucleatum* (green triangles), or *P. gingivalis* (yellow triangles) at an MOI of 50. Following treatment, the expression of the E-cadherin gene and surface protein levels were assessed by qPCR and flow cytometry, respectively. Panel (**A**): Gene expression relative to that of GAPDH calculated by the 2^−ΔCt^ method. Panel (**B**):—Percentage of Ca9-22 cells positive for E-cadherin (E-cadherin+ cells). Panel (**C**): Mean fluorescence intensities (MFI) of E-cadherin on positive CA9-22 cells. Each data point represents an individual experiment; lines and error bars indicate the mean and SD, respectively. Statistical significance (*p* < 0.05) is indicated as follows: *—significantly decreased percentage of E-cadherin+ cells or MFI after treatment with vitamin D_3_ metabolites. ^#^—significantly increased gene expression following vitamin D_3_ metabolite treatment. ^§^—significantly higher gene expression after treatment with 1,25(OH)_2_D_3_ compared to 25(OH)D_3_. ^†^—significantly decreased MFI of E-cadherin+ cells after treatment with 1,25(OH)_2_D_3_ compared to 25(OH)D_3_.

**Figure 6 ijms-26-07001-f006:**
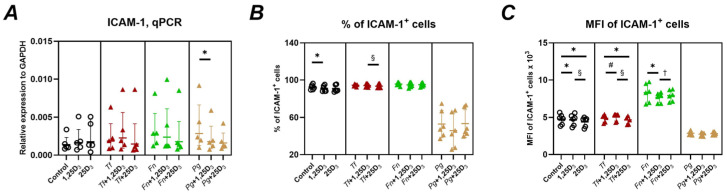
Effect of vitamin D_3_ metabolites on ICAM-1 gene and surface protein expression in Ca9-22 cells. Ca9-22 cells were treated with 10 nmol/L of either 1,25(OH)_2_D_3_ or 25(OH)D_3_ for 24 h. Treatments were performed on uninfected cells (circle) and cells infected with *T. forsythia* (red triangles), *F. nucleatum* (green triangles), or *P. gingivalis* (yellow triangles) at an MOI of 50. Following treatment, the production of ICAM-1 gene and surface protein was assessed by qPCR and flow cytometry, respectively. Panel (**A**): Gene expression relative to that of GAPDH calculated by the 2^−ΔCt^ method. Panel (**B**): Percentage of Ca9-22 cells positive for ICAM-1 (ICAM-1^+^ cells). Panel (**C**): Mean fluorescence intensities (MFI) of E-cadherin on positive CA9-22 cells. Each data point represents an individual experiment; lines and error bars indicate the mean and SD, respectively. Statistical significance (*p* < 0.05) is indicated as follows: *—significantly decreased ICAM-1 gene expression, percentage of ICAM-1^+^ cells or MFI of ICAM-1^+^ cells after vitamin D_3_ metabolite treatment. ^#^—significantly increased MFI of ICAM-1^+^ cells following vitamin D_3_ metabolite treatment. ^§^—significantly higher percentage of ICAM-1^+^ cells or MFI of ICAM-1^+^ cells after treatment with 1,25(OH)_2_D_3_ compared to 25(OH)D_3_. ^†^—significantly lower percentage of ICAM-1^+^ cells or MFI of ICAM-1^+^ cells after treatment with 1,25(OH)_2_D_3_ compared to 25(OH)D_3_.

## Data Availability

The original contributions presented in this study are included in the article. Further inquiries can be directed to the corresponding author(s).

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
