# Peer review of "Vitamin D3 Modulates Inflammatory and Antimicrobial Responses in Oral Epithelial Cells Exposed to Periodontitis-Associated Bacteria"

_ijms, 2025, doi:10.3390/ijms26147001_

Round 1

Reviewer 1 Report

Comments and Suggestions for Authors

The study is interesting, well-written and well-presented. Understanding the role of vitamin D3 is important in management of periodontitis. I have the following comments.

I suggest changing the title as there is no specific hypothesis provided on how vitamin D3 may shape the host pathogen interaction. The results only report some observations in relation to inflammatory mediators and antimicrobial peptide without building specific model about such interaction.

What was the reason for testing Tannerella forsythia? Please provide a brief rationale.

How do authors interpret the difference in IL-1β gene expression among the three tested organisms?

Authors should explain limitations of the experimental model used and how this can affect interpretation of the study findings.

Some references require update.

Author Response

Please find our response in the enclosed document.

Reviewer 2 Report

Comments and Suggestions for Authors

Karaca and coworkers present a really interesting study on the effects of Vitamin D 25 and 1,25.  Overall, the paper has merit and I believe it should be published after addressing some major important points. To be very specific, I will address these points according to the line numbers in the version I see.

The title sounds really broad.  Sounds like it could be the title of a review paper. Consider a title that is more specific to your findings and main message.

The abstract sounds great.

Introduction:

lines 56-59 many bacterial taxa are mentioned, but although these seem to be important in onset of periodontitis, these taxa are not relevant to the rest of the paper. If you removed them, the reader would not miss them.  On the other hand, as part of the multispecies community, and with a big role yet defined, are the commensal streptococci. The whole paper is testing three red-complex members, but never as a group, and never with commensals as it would occur in vivo. This would also become a point  that was never included in the discussion.

line 66: The VDR is mentioned, but never explored any further in terms of signaling.  Since VD3 is fat-soluble, it would easily cross the cell membrane. Is the VDR an intracellular protein? What do we know about its signalling? if nothing, you may say that this is also unexplored.

The paper finally mentions the physiological levels of VD3 at around 10 nM in the discussion. Please include this information in the introduction to guide the readers in the methods section for such concentration. Is the [VD3] in the gingival crevicular fluid known?  If so, include it.

Lines 103 - 105: GECs express the enzyme that converts 25(OH)D3 to 1,25(OH)D3.  Do you know if your cell line Ca9-22 also produces such enzyme?  Because this cell line derived from an oral squamous cell carcinoma, it easily raises the question. Based on this, I will also request the authors to include a fair justification for the use of cancer cell line. There are oral cell line models that derive from healthy tissue and could yield different results. 

After the above issues, I think the authors did a good job arriving to the scientific questions/gaps, aims to contribute in this field and significance (lines 111 - 119).

Methods:

Lines 407 - 410. I work with Fn and Pg and I always add 1 ug/mL menadione (vitamin K) to the agar, otherwise these not grow.  I use Pg 33277 as well.  If you do add menadione, please state it.  Otherwise, explain how Pg and Fn can grow without this reagent.

Line 413. Why are the cells cultured in MEM without FBS?  How does this help the experiment? Please explain.

Lines 416 - 417: Are the bacteria grown in broth at all?  are they simply resuspended from agar colonies in MEM?  Does the MEM contain antibiotics? if not, mention it is antibiotic-free.  If yes, are the bacteria dying?

Lines 419 - 420: What is the OD that yields an MOI of 50?  I wouldn't be able to replicate the experiment without this information.

Lines 421 - 422: Sounds like the bacteria and/or the VD3 stay on the cells for 24 hours. So there are no washes in between to remove anything?  Please clarify.

For every qPCR experiment, we need to see the variability of the control. In other words, once you calculate the delta Ct of the control (Ct target - Ct GAPDH), take the average of these values.  Then, take the delta delta Ct of each control against its own average. This will give you the variability in the control for each gene. Otherwise the control is always 1 no matter the n value, but we know it is virtually impossible to get exactly a zero on the delta delta Ct and 2^0 = 1 every single time. This is crucial when running tests of significance and deciding what is statistically significant and what is not.  Major point to correct. Perhaps your interpretations of these results will also change.

Results:

For all the results, state them in present form.  Example, "...after treatment with 1,25, the gene was significantly upregulated..." should be "...after treatment with 1,25, the gene is significantly upregulated..."

Lines 128 - 131: Treatment with Pg and 1,25 seems to me that increases IL-1 beta protein compared to Pg and 25.  Please check this again.

The LL-37 data is the most significant in terms of gene expression. It looks like the bacteria not changing these results much. Since it is important and significant, please report the mean values of the LL-37 ELISA in line 162 and on.

The description for the results in Fig 3 are okay.

Fig 4 indicates the CFUs start at MOI = 50.  If the cells are at 170,000/well, the number of bacteria should be 8.5 x 10^6.  Going back to the methods for this experiment, where the bacteria resuspended in MEM, checked for OD = ____ and then pellet them down to wash them and then resuspend them?  Please explain the details of this method so other can replicate it. The controls for all three bacteria are below a million. Please explain this, or fix the methods so it makes more sense. Also, 24 hours is not enough time to grow colonies for Pg.  At least 48 hours should be allowed. I am not sure about the other two bacteria. Please double-check the protocols. Just like the LL-37 data, this is the other major punchline of the study and I want to make it's really well explained and reproducible by others.  

Lines 273 - 274: 1,25 increases I-CAM gene expression regardless of the bacteria, but you seem to think it's only when Pg is included. Check this again. 

Discussion:

Authors make a good points that their data is significant, yet not substantial. They also indicate a few contrasts or disagreeing data with other studies. What cell lines were used in those studies? In lines 363 - 370 make sure the reader knows these are cancer cells and not normal. In the limitations (lines 390 - 393), please include the fact that no commensal bacteria, or their metabolites, where included in this study. This is paramount to mimic the in vivo environment a bit closer.

This paper addresses a very important point within the field of host-bacteria interactions regarding periodontal disease. It definitely merits publication, but it needs to have all above corrections made.

All the best!

Author Response

(The authors gave the same response as above.)

Round 2

Reviewer 2 Report

Comments and Suggestions for Authors

Karaca and coworkers have made important improvements on their manuscript. I am pleased with many of the edits. Still, a few more important points are still in need of attention:

Authors mentioned: "Despite the transcriptional up-regulation, IL-1β protein levels in the culture supernatants were significantly reduced following 1,25(OH)â‚‚D₃ treatment in cells infected with F. nucleatum and P. gingivalis, compared to untreated or 25(OH)D₃-treated controls (Fig. 1B)."  For Tf and Fn infections, that is correct. However, I see higher IL-1β with Pg and 1,25(OH)â‚‚D₃ treatment than Pg alone and than Pg with 25(OH)D₃ (Fig. 1B). Please take a look at this again and rephrase.

Authors wrote: "At the protein level, LL-37 levels in the conditioned media were also significantly elevated by both metabolites (Fig. 2B), although 25(OH)D₃ led to higher LL-37 peptide concentrations (14-73 ng/ml) than 1,25(OH)â‚‚D₃ (4-24 ng/ml)."  Which conditions are 14-73 ng/mL and 4-24 ng/mL?  Some of these data points are in the hundreds of ng/mL.  Please specify.

The y axis on Figures 4A and B, I assume, are shown in the log base 10 scale, where 5 = 10^5 CFUs  and 6 = 10^6 CFUs and so on. This is not clear. 

If my assumption on the log scale is correct, my previous question still stands. The MOI = 50 is based on 1.7 x 10^5 cells/well x 50 = 8.5 x 10^6 bacteria. If the y axis is on the log base 10, a lot of the bacteria are missing even in the control spent media.  In the case of 4B, this is even more alarming because the bacteria are suspended in media alone (no vit D metabolites) and the final values are below the 10^6 mark. Please indicate somewhere (results or discussion) the loss of so much bacteria from the controls of these experiments.  

Lastly, on Figure 4 A and B, once my above point has been satisfied, it might be helpful to the reader to see a figure 4C where you show a % of live bacteria in the cell culture spent media (4A) compared to fresh media with and without vit D metabolites (4B).  This would drive home the point that the cells + vit D metabolites are producing antimicrobials that kill the bacteria.

All the best!

Author Response

Thank you for reevaluating our manuscript and providing insightful comments. Our response is in the enclosed file.
